# A New Investigation into the Molecular Mechanism of Andrographolide towards Reducing Cytokine Storm

**DOI:** 10.3390/molecules27144555

**Published:** 2022-07-17

**Authors:** Abdulaziz Alzahrani

**Affiliations:** Pharmaceuticals Chemistry Department, Faculty of Clinical Pharmacy, Al Baha University, Al Baha 65779, Saudi Arabia; alzahraniaar@bu.edu.sa

**Keywords:** andrographolide, cytokine, interleukin

## Abstract

Cytokine storm is a condition in which the immune system produces an excessive number of inflammatory signals, which can result in organ failure and death. It is also known as cytokine release syndrome, CRS, or simply cytokine storm, and it has received a lot of attention recently because of the COVID-19 pandemic. It appears to be one of the reasons why some people experience life-threatening symptoms from COVID-19, a medical condition induced by SARS-CoV-2 infection. In situations where natural substances can be exploited as therapeutics to reduce cytokine storm, the drug development process has come to the rescue. In the present study, we tested the potentiality of Andrographolide, labdane diterpenoid targeting several key cytokines that are secreted as a result of cytokine storm. We used molecular docking analyses, molecular dynamics simulations, and pharmacokinetic properties to test the stability of the complexes. The compound’s binding energy with some cytokines was over −6.5 Kcal/mol. Furthermore, a post-molecular dynamics (MD) study revealed that Andrographolide was extremely stable with these cytokines. The compound’s pharmacokinetic measurements demonstrated excellent properties in terms of adsorption, distribution, metabolism, and excretion. Our research revealed that this compound may be effective in lowering cytokine storm and treating severe symptoms.

## 1. Introduction

Cytokines are small proteins secreted by cells that have a specific effect on cell contacts and communication [1]. They are also termed lymphokine cytokines, which are produced by lymphocytes (monokine is a kind of cytokine produced mostly by monocytes and macrophages, chemokine make guiding movement of leukocytes, and interleukin cytokines are created by interleukin cytokines made by one leukocyte acting on other leukocytes) [1]. In addition, cytokines classified as mononuclear cells, such as macrophages, have been referred to as monokines, while the cytokines produced by activated T lymphocytes are termed lymphokines. Cytokines can be classified according to the split of labor: T-helper-1 of cytokines such as interleukin (IL)-2, IL-12, TNF-a, IFN-γ; T-helper-2 of cytokines such as IL-3, IL-4, IL-5, IL-13; and T-helper-3 of cytokines such as IL-10, TGF-β. Chemokines such as CCL2 and CXCL10 are small cytokines, or called signaling proteins secreted by cells. They have the ability to induce directed chemotaxis in nearby responsive cells; they are chemotactic cytokines. They are secreted as a result of inflammatory response. The immune system has a critical role in the control and resolution of acute viral diseases such as severe acute respiratory syndrome coronavirus 2 (SARS-CoV-2) infection; hence, it is important to understand cytokine storm. Immunopathogenesis, or an out-of-control immunological response to a disease, can be catastrophic, producing severe inflammation and even death [2]. The immune system is stimulated and cells are recruited in the majority of SARS-CoV-2-infected people, clearing the virus in the lungs. After that, the immunological response progressively fades and the patients recover. However, in some cases, a dysregulated immune response is noted, resulting in a huge release of cytokines and chemokines, known as a ‘cytokine storm,’ which mediates extensive inflammation in the lungs [3].

According to studies, cytokine storm is linked to the advancement of SARS-CoV-2 infection; a higher level of cytokine storm is linked to a more severe disease development [4]. In patients with COVID-19, plasma levels of cytokines IL-1, IL-1RA, IL-7, IL-9, IL-10, FGF, G-CSF, GM-CSF, PDGF, VEGF, IFN, and TNF, and chemokines CXCL8, IP10, MCP1, MIP1, and MIP1 are much higher than in healthy people. Furthermore, chemokines IP10, MCP1, and MIP1 and pro-inflammatory cytokines IL-2, IL-7, IL-10, G-CSF, and TNF are higher in severe patients compared to mildly infected patients [4].

Andrographolide is a labdane diterpenoid isolated from Andrographis paniculate, a medicinal herb. It has anti-inflammatory, anti-allergic, anti-platelet aggregation, antineoplastic, anti-HIV, and hepatoprotective qualities [5]. Andrographolide is a strong immunomodulator that has been shown to increase immunological response, control NK cell and cytokine production, and stimulate the generation of cytotoxic T lymphocytes [6]. In LPS/IL-4-activated murine macrophages, andrographolide reduced the levels of inflammatory cytokines TNF, IL-12, IL-1, IL-6, and IL-18 in a dose-dependent manner [5]. After pMCAO activation, andrographolide inhibits inflammatory mediators IL-1, TNF, prostaglandin E2 (PGE2), NADPH oxidase 2 (NOX2), and inducible nitric oxide synthase (iNOS) in ischemic brain regions [7]. Andrographolide therapy reduced the levels of pro-inflammatory cytokines TNF, IL-1, and IL-6, as well as their related mRNA expression levels, in LPS-stimulated RAW264.7 cells. The andrographolides, which have shown anti-inflammatory and immune-stimulatory properties, inhibit activation factors mediated by inflammatory response and also reduce expression of pro-inflammatory proteins [8]. In a previous study of in vivo anti-inflammatories, it was shown that TNF-α and GM-CSF release from mouse peritoneal macrophages were inhibited by andrographolide [9]. It has also been described to suppress IL-2 production and T-cell proliferation in a mixed lymphocyte reaction and to inhibit dendritic cell maturation and antigen presentation [10].

The drug design and discovery driven by advances in powerful cheminformatics and various in silico tools culminated in the revolution of the ‘bioinformatics era’ [11]. The in silico approach uses a combination of concepts from computer algorithms and statistical methods with advancements in high-performance computation in biological science [12]. Therefore, bioinformatics and computational biology has become the central paradigm in speeding up the process and reducing the cost of probing new potential candidates for any targeted disease [13]. Two major approaches, i.e., structure- and ligand-based virtual screening approaches, have been used extensively to discover lead compounds despite having some limitations in them [14]. Integration of these in silico techniques seems to be a reliable approach in identifying potential candidates for improved therapeutic uses.

Taking advantage of these techniques, we selected Andrographolide to identify its potential efficacy against eighteen cytokines in this study. Their classification, main sources, receptor, target cell and major function are listed in Table 1 [15].

## 2. Materials and Methods

### 2.1. Preparation of Compound and Receptors

Eighteen cytokines named IL-2, IL-3, IL-4, IL-5, IL-6, IL-13, IFN-γ, IL-8, IL-10, IL-12, IL-17, G-GSF, GM-CSF, IP10, M-CSF, MiP1α, MiP1β, and TNF-α were selected for this study. The crystal structures of the cytokines were collected from the Research Collaboratory for Structural Bioinformatics (RCSB) Protein Data Bank (PDB) [16]. All the ligand molecules, hetero atoms, and the water molecules of these structures were removed from the structures using Discovery Studio 4.0 client (http://accelrys.com/products/discovery-studio/, accessed on 20 June 2022) [17]. Later, the 3D structure of Andrographolide (Figure 1) was collected from the PubChem database and prepared for molecular docking.

### 2.2. Molecular Docking Simulation of Andrographolide with Cytokines

Molecular docking was performed using PyRX [18], an open-source virtual screening software to determine receptor–ligand interactions. The important ligand binding site residues were known from the CASTp server (https://sts.bioe.uic.edu/castp/, accessed on 20 June 2022) and were enclosed within the suitable parameters of grid box dimensions. AutoDock Vina (ADV) [19], implemented in PyRX, was used to accomplish all the docking simulations within the predetermined parameters. Further, the protein–ligand interaction was visualized by Discovery studio 4.0 client and Pymol 1.1 [20]. To validate the software efficiency, we selected compound 52 and reparixin, which are known to inhibit IL-4 and IL-8, respectively, and performed molecular docking as controls. [21,22]

### 2.3. Molecular Dynamics (MD) Simulation Analysis

Molecular dynamics simulation of the four protein-ligand complexes was performed using GROMACS 2021.1 the University of Groningen, city of Groningen, Northern Netherlands [23] version and Linux 5.4 package. The GROMOS96 54a7 [24] force field was selected as the force field for the cytokines and the ligand topologies were generated from the PRODRG the University of Dundee, city of Dundee, UK [25] server. All the complexes were solvated using simple point charge (SPC) water molecules in a rectangular box. To make the simulation system electrically neutral, this required a number of Na^+^ and Cl^−^ ions, which were added while 0.15 mol/L salt concentrations were set in all the systems. Using the steepest descent method, all the solvated systems were subjected to energy minimization for 5000 steps. Afterwards, the NVT (constant number of particles, volume, and temperature) series, NPT (constant number of particles, pressure, and temperature) series, and the production run were conducted in the MD simulation [26]. The NVT and the NPT series were conducted at a 300 K temperature and 1 atm pressure for the duration of 300 ps. The V-rescale thermostat and Parrinello–Rahman barostat were selected for the performed simulation. Finally, the production run was performed at 300 K for a duration of 100 ns (nanoseconds). Thereafter, a comparative analysis was performed measuring the root-mean-square deviation (RMSD), root-mean-square fluctuation (RMSF), radius of gyration (Rg), solvent-accessible surface area (SASA), and hydrogen bonds to analyze their stability. The Xmgrace [27] program was used to represent the analyses in the form of plots.

### 2.4. ADME Analysis

The SwissADME [28] online tool (http://www.swissadme.ch, accessed on 20 June 2022) was utilized to evaluate the pharmacokinetic properties of Andrographolide.

## 3. Results

### 3.1. Molecular Docking Analysis of Andrographolide with the Cytokines

AutoDock Vina predicted nine possible binding positions for each compound as output. The best position was chosen for each compound based on the lowest docking energy. The docking energy score of all the cytokines with Andrographolide (Figure 1) ranged from −1.1 to −7.5 Kcal/mol. The amino acid interactions of Andrographolide with the cytokines were also identified. IL-3, IL-10 and G-GSF showed a higher docking energy of >7.0 Kcal/mol with Andrographolide. The highest number of interactions (8) was found in an IFN-γ—Andrographolide complex, while IL-6 and MiP1B established the highest number of hydrogen bonds with the compound. The docking results and their interactions with Andrographolide are shown in Table 2 and Figure 2 and Figure 3, while the docking results of two cytokines with the controls are shown in Table 3, and the interactions are shown in the Appendix A.

### 3.2. Molecular Dynamics (MD) Simulation Results

The dynamic movements of atoms and conformational variations of Cα atoms of the protein–ligand complexes were calculated by RMSD to detect their stability. It is observed that except for the IL5, all the complexes exhibit lower RMSD (<1.0 nm) with no major fluctuations, indicating their greater stability shown in Figure 4. The flexibility of each residue was calculated in terms of RMSF to gain better insight on the region of cytokines that are being fluctuated during the simulation. It can be understood that the binding of compounds makes the cytokines slightly flexible in several areas (Figure 4). The compactness of the complex was represented by the radius of gyration (Rg). The lower degree of fluctuation throughout the simulation period indicates the greater compactness of a system. The Rg of the complexes were found to be similar, while the Andrographolide-IL10 complex was found to be slightly variant (Figure 4). Hydrogen bonding between a protein–ligand complex is essential to stabilize the structure. It was observed that the highest number of conformations of the protein formed up to six hydrogen bonds with Andrographolide (Figure 4). Interaction between protein–ligand complexes and solvents was measured by the solvent-accessible surface area (SASA) over the simulation period. Therefore, the SASA of the complex was calculated to analyze the extent of the conformational changes that occurred during the interaction. Interestingly, all the cytokines featured a slight reduction in the surface area, showing a relatively lower SASA value than the starting period (Figure 4).

### 3.3. ADME Analysis

The ADME properties of Andrographolide were predicted using a Swiss ADME server (Table 4). In many attributes, the compound showed significant values. Interestingly, it does not violate Lipinski’s rule of five and has high GI absorption. The compound does not inhibit any isoform of the cytochrome P450 enzyme. The log k_p_ (skin permeation) was in a good range and it also had efficacy to cross the blood–brain barrier. A PAINS alert confirmed the absence of any catechol moieties in the compound and synthetic accessibility values were also found to be low.

## 4. Discussion

Apart from traditional methods, computer-assisted drug design (CADD) approaches are now used efficiently in the drug discovery and development process. Molecular docking and molecular dynamics simulations are commonly used to perform the virtual screening of compounds to examine how ligands inhibit a target and confirm their stability towards specific target proteins, among numerous approaches to CADD that are recognized as promising techniques. Additionally, the rising development of robust computational techniques allows for the assessment of the pharmacological properties of drugs prior to engaging in an experimental trial. As a result, in this study, rigorous CADD methodologies were used to examine the drug capabilities of Andrographolide against a range of cytokines.

Considering the significance of cytokines and chemokines in immune system mediation maintaining anti-viral immunity, we tested Andrographolide’s efficacy against different cytokines by looking at both its binding affinity and ADMET characteristics. Andrographolide was tested against seventeen cytokines for this purpose. When binding with IL3, the compound had the maximum binding affinity of −7.5 kcal/mol and interacted with six amino acid residues in the cytokines’ active site. We further investigated their stability with Andrographolide, conducting MD simulation for a 100 ns time period, even though they had strong binding affinity with other cytokines as well.

The compound was further used for molecular docking analysis to determine their binding affinities towards different cytokines. The result of the analysis indicated good binding energies of most cytokines ranging between −5.3 and −7.5 kcal/mol, better than the minimum binding energies of MiP1α (−1.1 kcal/mol), IP10 (−4.2 kcal/mol), IL-8 (−3.8 kcal/mol), and IL-2 (−4.3 kcal/mol) (Table 2). The compound formed stable interactions with several of the active site residues of the target cytokines. Almost all the cytokines were stabilized by at least one hydrogen bonding interaction and several other interactions such as carbon–hydrogen bonding and Pi-Alkyl interactions.

The stability of Andrographolide with the respected cytokines was confirmed by the MD simulation findings of the top 11 complexes. The RMSD plot demonstrated that all of the complexes are stable, and the RMSD values did not fluctuate significantly over time. The complexes did not show significant variability and were within an acceptable range, according to the RMSF analysis. The radius of gyration (Rg) data revealed that each complex had a substantially comparable compactness characteristic. The SASA data indicated that the volume of the complexes had dropped somewhat. Throughout the simulation, a significant number of hydrogen bonds were identified in all the complexes, which once again explained their conformational stability. As a result, Andrographolide can bind to these cytokines, interfering with their action. Thus, by inhibiting these cytokines, several functions such as the production of granulocyte and monocyte, and the proliferation of B and T cells can be altered, thereby reducing the cytokine storm.

Andrographolide has a high gastrointestinal absorption potency, with a skin permeability coefficient (Log Kp) of −6.90, according to its ADME characteristics. It does not break Lipinski’s rule of five; therefore, it might readily be mimicked by other natural cytokine ligands. Furthermore, the permeability of the blood–brain barrier membrane revealed that the compound is unable to cross the blood–brain barrier. Pan-assay interference chemicals (PAINS) are chemical compounds that react nonspecifically with a variety of biological targets, resulting in false positive results in high-throughput screens. It does not have any catechol groups, according to the PAINS filter. It has a low synthetic accessibility score, indicating that the compound can considerably more easily be produced in experimental laboratories.

There may be some limitations in this study because it was conducted via in silico analysis. Despite the positive results with the cytokines, the compound has yet to be tested in an animal model. To assure its therapeutic efficacy, sufficient experimental validations are required.

## 5. Conclusions

Our study proposed and tested a new compound, Andrographolide, against several crucial cytokines involved in cytokine storm. Binding of the compound to the target cytokines may eventually create an alteration in entering in their action and producing a lesser amount. The compound showed better results in both binding with the proteins in efficient oral bioavailability and less toxicity. Considering binding affinities, interactions with target proteins, and ADME profile, we have analyzed and validated the stability of protein–ligand complexes using an MD simulation approach and found intriguing results for all of them. The results of this study might be valuable references for further experimental research in designing new potent cytokine regulators. Hence, it can be concluded, after further in vitro and in vivo experiments, that Andrographolide can be used as a therapeutic against cytokine storm.

## Figures and Tables

**Figure 1 molecules-27-04555-f001:**
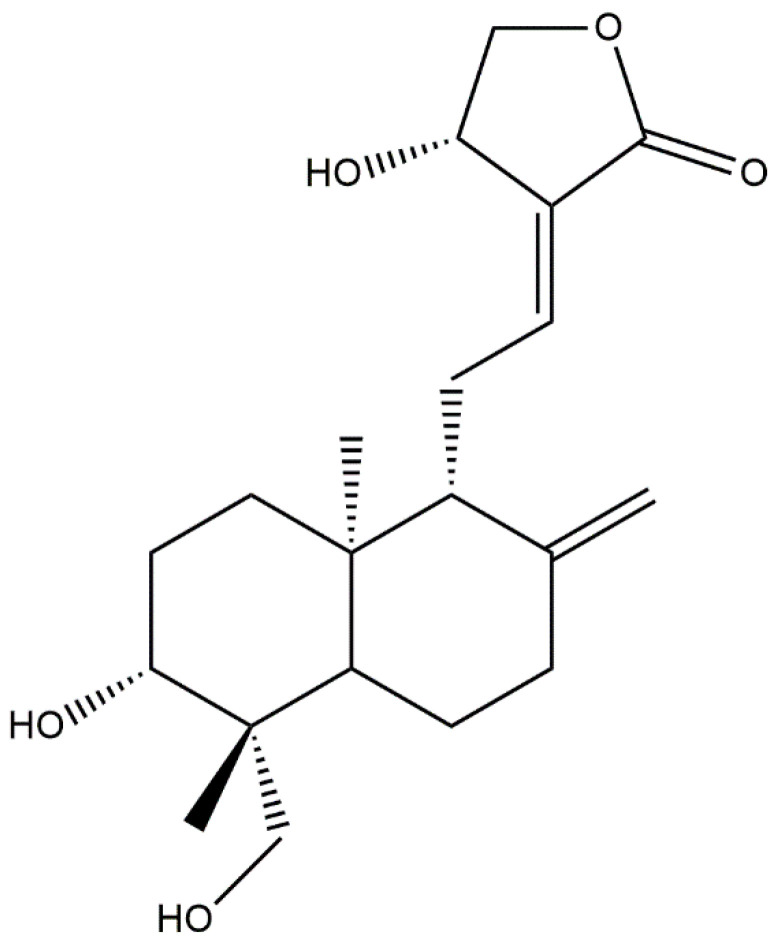
Crystal structure of Andrographolide.

**Figure 2 molecules-27-04555-f002:**
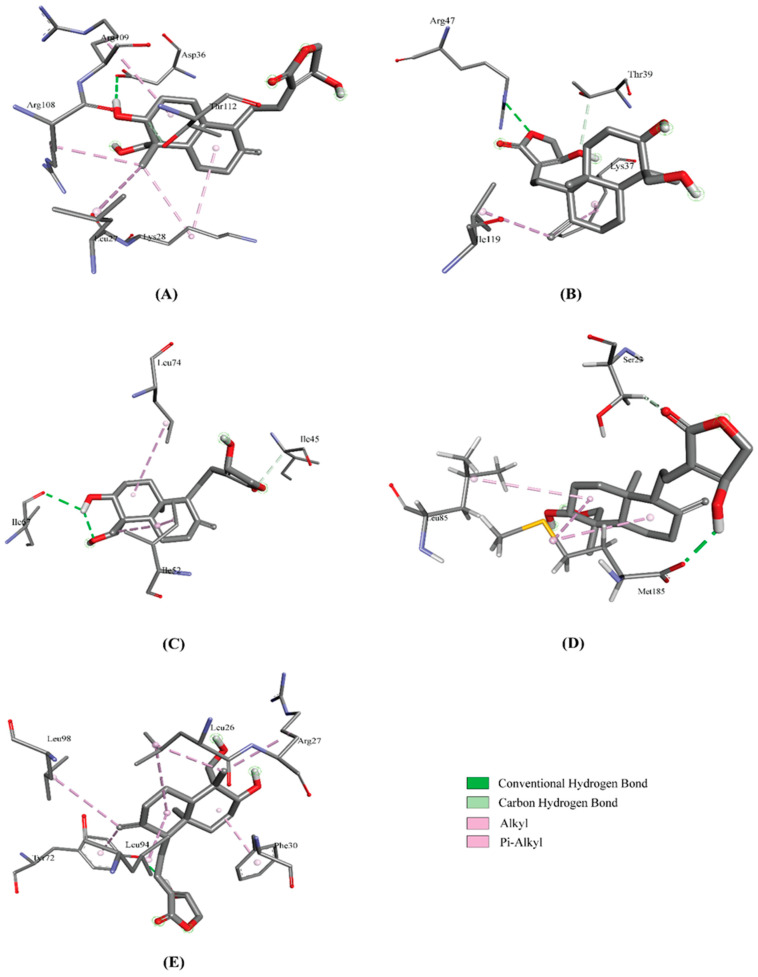
Interactions of IL-3 (**A**), IL-4 (**B**), IL-5 (**C**), IL-6 (**D**) and IL-10 (**E**) with Andrographolide.

**Figure 3 molecules-27-04555-f003:**
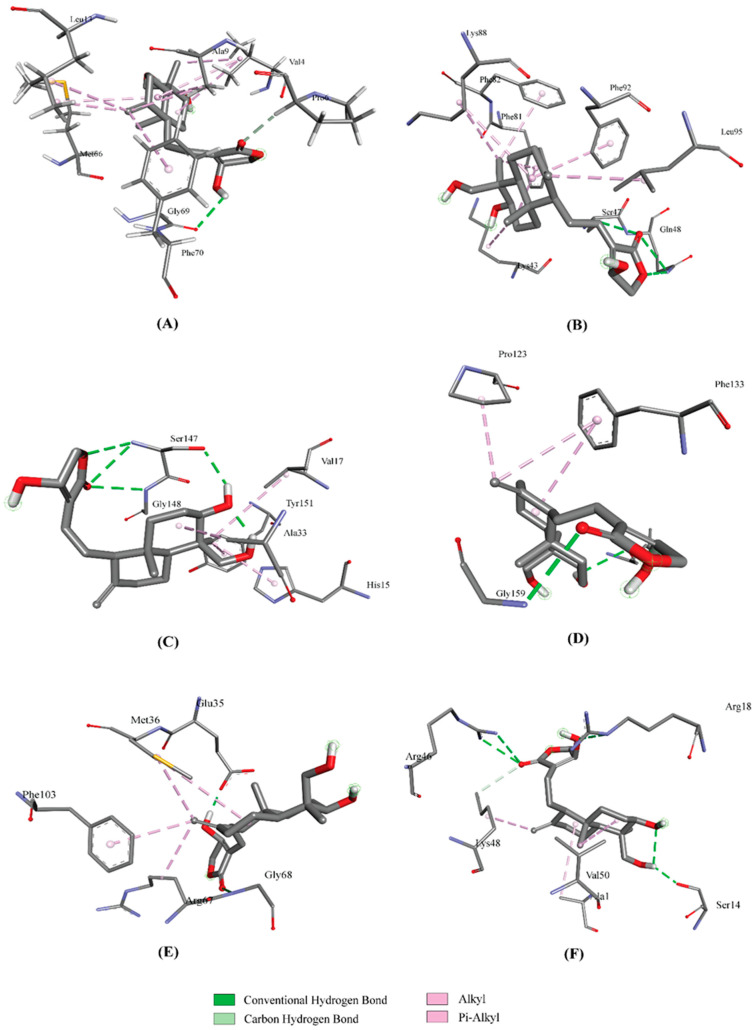
Interaction of IL-13 (**A**), IFN-γ (**B**), TNF-α (**C**), G-CSF (**D**), GM-CSF (**E**), and MiP1β (**F**) with Andrographolide.

**Figure 4 molecules-27-04555-f004:**
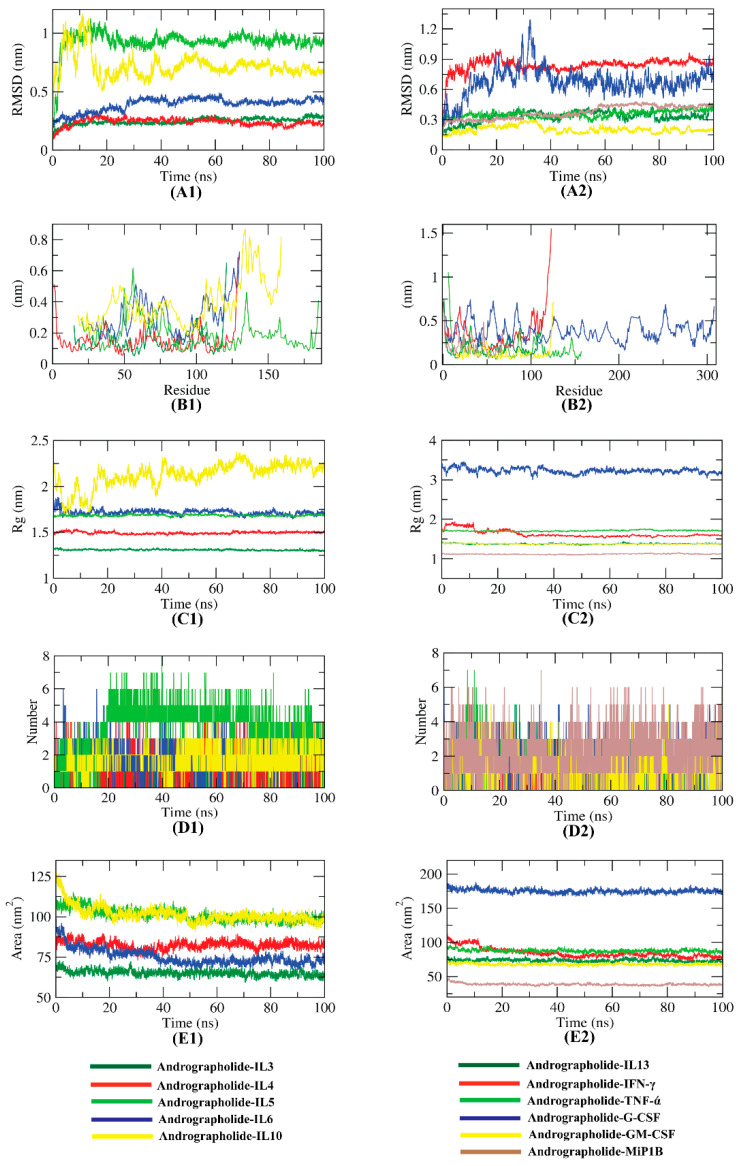
RMSD (**A1**,**A2**), RMSF (**B1**,**B2**), Rg (**C1**,**C2**), hydrogen bond (**D1**,**D2**), and SASA (**E1**,**E2**) results of the Andrographolide–cytokine complexes after 100 ns simulation.

**Table 1 molecules-27-04555-t001:** List of selected cytokines and their classification and roles.

Cytokine	Classification	Origin	Receptor	Aim Cell	Role
G-CSF	Pro-inflammatory	Fibroblasts, endothelium	CD114	Stem cells in BM	Granulocyte production
GM-CSF	Adaptive immunity	T cells, macrophages, fibroblasts	CD116, CDw131	Stem cells	Growth and differentiation of monocytes, and eosinophil, granulocytes production
M-CSF	Adaptive immunity	Fibroblasts, endothelium	CD115	Stem cells	Monocyte production and activation
IL-2	Adaptive immunity	Th1 cells	CD25	Activated T and B cells, NK cells	Proliferation of B cells, activated T cells, NK cell function
IL-3	Adaptive immunity	T cells	CD123, CDw131	Stem cells	Hematopoietic precursor proliferation and differentiation
IL-4	Adaptive immunity	Th Cells	CD124	B cell, T cell, macrophages	Proliferation of B and cytotoxic T cells, enhances MHC class II expression, stimulates IgG and IgE production
IL-5	Adaptive immunity	Th2 Cells and mast cells	CDw125, 131	Eosinophils, B-cells	B-cell proliferation and maturation, stimulates IgA and IgM production
IL-6	Pro-inflammatory	Th Cells, macrophages, fibroblasts	CD126, 130	B-cells, plasma cells	B-cell differentiation
IL-8	Pro-inflammatory	Macrophages	IL-8R	Neutrophils	Chemotaxis for neutrophils and T cells
IL-10	Anti-inflammatory	T cells, B cells, macrophages	CDw210	B cells, macrophages	Inhibits cytokine production and mononuclear cell function
IL-12	Anti-inflammatory	T cells, macrophages, monocytes	CD212	NK cells, macrophages, tumor cells	Activates NK cells, phagocyte cell activation, endotoxic shock, tumor cytotoxicity, cachexia
IL-13	Anti-inflammatory	Th2 Cells, mast cells, eosinophils, basophile and nunocytes	IL13Rα1	Act on monocyte, fibroblast and B cell growth	Regulate eosinophilic inflammation and mucosa secretion
IL-17	Pro-inflammatory	Th17 cells	IL-17R	Monocytes, neutrophils	Recruit monocytes and neutrophils to the site of infection. Activation of IL-17 in turn activate downstream of many cytokines and chemokine, such as IL-1, IL-6, IL-8, IL-21, TNF-β, and MCP-1
IFN-γ	Pro-inflammatory	T Cells and NK cells	CDw119 (IFNG R1)	Various	Anti-viral, macrophage activation, increases neutrophil and monocyte function, MHC-I and -II expression on cells
TNF-α	Pro-inflammatory	Macrophages	CD120a, b	Macrophages	Phagocyte cell activation, endotoxic shock
MIP-1α	Pro-inflammatory	Macrophages and monocytes		Hematopoietic cells	Immune responses towards infection and inflammation
MIP-1β	Pro-inflammatory	Macrophages and monocytes		Hematopoietic cells	Immune responses towards infection and inflammation
IP 10	Pro-inflammatory	Monocytes, Endothelial cells and Fibroblasts			Antitumor activity, and inhibition of bone marrow colony formation and angiogenesis

**Table 2 molecules-27-04555-t002:** Molecular docking results of different cytokines with Andrographolide.

Cytokines	PDB Code	Binding Affinity (Kcal/mol) with Andrographolide
IL-2	4NEJ	−4.3
IL-3	5UWC	−7.5
IL-4	2B8U	−6.4
IL-5	3QT2	−6.3
IL-6	1ALU	−6.0
IL-13	3BPO	−6.7
IFN-γ	6E3L	−6.2
IL-8	1IL8	−3.8
IL-10	1ILK	−7.1
IL-12	1F45	−5.3
IL-17	6HGO	−6.2
G-GSF	3UEZ	−7.2
GM-CSF	2GMF	−6.2
IP10	1O7Y	−4.2
M-CSF	1HMC	−6.0
MiP1α	2X6G	−1.1
MiP1β	4RAL	−6.3
TNF-α	2AZ5	−6.0

**Table 3 molecules-27-04555-t003:** Molecular docking results of cytokines with known inhibitors.

Cytokines	Binding Affinity (Kcal/mol)
IL-4-Compound 52	−5.3
IL-8-Reparixin	−5.8

**Table 4 molecules-27-04555-t004:** Important ADME properties of Andrographolide.

Parameters	Cytisine
Num. H-bond acceptors	5
Num. H-bond donors	3
Molecular weight (g/mol)	350.45
Lipinski violation	0 violation
GI absorption	High
BBB permeant	No
CYP1A2 inhibitor	No
CYP2C19 inhibitor	No
CYP2C9 inhibitor	No
CYP2D6 inhibitor	No
CYP3A4 inhibitor	No
Log K_p_ (skin permeation) (cm/s)	−6.90
PAINS	0 alert
Brenk	2 alerts: isolated_alkene, michael_acceptor_1
Synthetic accessibility	5.06

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
