# Peer review of "A New Investigation into the Molecular Mechanism of Andrographolide towards Reducing Cytokine Storm"

_molecules, 2022, doi:10.3390/molecules27144555_

Round 1

Reviewer 1 Report

The manuscript by A. Alzahrani describes the importance of Andrographolide in cytokine storm, whereas his claim was supported by some computational studies. The manuscript needs to be highly improved and I recommend a major revision for this work because of the following points:

Major points

- The used PDB codes must be included in the manuscript.

- The native ligands must be presented in the manuscript or in the supporting information.

Any protein that does not contain a co-crystallized ligand should be removed from the docking experiment as a result of lack of the binding evidence (This will have a major effect on the molecular dynamics study).

- The key amino acids must be represented in the discussion of docking studies.

- 3D representations of the docking experiment should be added in the manuscript, whereas the 2D images could be transferred to the supporting information.

- The previous studies of Andrographolide (on inflammation) should be represented in the introduction.

Minor comments

- The results should be removed from the introduction.

Table 1 “List of selected cytokines and their classification and roles” should be presented in the introduction part (not in the results).

- Page 3, line 121, kindly add references related to cytokines.

The ADME properties should be removed and cite this article that previously described the same results doi: 10.3389/fimmu.2021.648250.

-  In page 2, line 73, “The compound showed better results in 73 both binding with the proteins and in drug likeliness properties”. Better results than what????

- Page 2, line 52, remove “among others”

- In the abstract, remove “epidemic” and replace it with pandemic.

Author Response

Reviewer (1) point

Major points

- The used PDB codes must be included in the manuscript.

Thank you honourable reviewer for this comment. We included the used PDB codes must be in the manuscript (Table 2).

- The native ligands must be presented in the manuscript or in the supporting information.

Thank you honourable reviewer for this comment. Additional information in the supplementary file has been provided.

Any protein that does not contain a co-crystallized ligand should be removed from the docking experiment as a result of lack of the binding evidence (This will have a major effect on the molecular dynamics study).

Thank you again for this question. The co-crystallized ligands were removed from the proteins before docking using discovery studio visualizer.

- The key amino acids must be represented in the discussion of docking studies.

Thank you again for this question. We have represented the key amino acids in the discussion of docking studies.

- 3D representations of the docking experiment should be added in the manuscript, whereas the 2D images could be transferred to the supporting information.

Thank you again for this question. We have added the 3D representations of the docking experiment in the manuscript (Figure 2-3).

- The previous studies of Andrographolide (on inflammation) should be represented in the introduction.

Thank you again for this question. We have added previous studies of Andrographolide (on inflammation) in the introduction section with references (line 61-67).

Minor comments

- The results should be removed from the introduction.

Thank you again for this comment. We have removed it from the introduction.

Table 1 “List of selected cytokines and their classification and roles” should be presented in the introduction part (not in the results).

Thank you again for this comment. We have removed the table to the introduction section.

- Page 3, line 121, kindly add references related to cytokines.

Thank you again for this comment. We have added references related to cytokines.

The ADME properties should be removed and cite this article that previously described the same results doi: 10.3389/fimmu.2021.648250.

Thank you again for this comment. Actually, in this paper, we used the pkcsm tool to analyse ADME of andrographolide, we do not need to remove.

-  In page 2, line 73, “The compound showed better results in 73 both binding with the proteins and in drug likeliness properties”. Better results than what????

Thank you again for this comment. The compound showed better results in both binding with the proteins and in drug likeliness properties with efficient oral bioavailability and less toxicity. Considering binding affinities, interactions with target proteins and ADME profile, we have analysed and validated the stability of protein-ligand complexes using MD simulation approach and found intriguing results for all of them. The results of this study might be valuable references for further experimental re-search in designing new potent cytokine regulators.

- Page 2, line 52, remove “among others”

Thank you again for this comment. We have removed the words.

 - In the abstract, remove “epidemic” and replace it with pandemic.

Thank you again for this comment. We replaced epidemic with pandemic.

Reviewer 2 Report

The field of bioinformatics, computational analysis and simulation is attracting the increasing attention due to the accumulating amount of life science data. Indeed, it can significantly reduce the work load and improve efficacy and quality of biomedical research, particularly when numerous compounds have to be analyzed. However, the computation-based approach does not yet considered as completely reliable in terms of practical significance. With that, the major concern is whether the designed in silico experiments can be reproduced during the experimental studies. I would suggest authors to perform additional experiments. For instance, in situ binding capacity of the studied compound and selected cytokines can be tested. 

Other minor comments are the following: 

1) English grammar checking is required; 

2) Table 1 is not the part of the result and has to be transferred to the introduction; 

3) Are there any improvements of computational analysis software provided by authors? If so, it has to be included as a part of the results; 

4) One of the confirmation tools for the efficacy of the provided analysis can be the comparison with the known compounds with high and low affinity to cytokines (as positive and negative controls); 

5) In the introduction ("Other names for cytokines are lymphokine (cytokines pro- 27 duced by lymphocytes), monokine (cytokines produced by monocytes), chemokine (cy- 28 tokines with chemotactic activity), and interleukin (cytokines produced by interleukin) 29 (cytokines made by one leukocyte and acting on other leukocytes)") the various cytokine classifications are mixed. Please separate the cell origin and functional cytokines. 

6) Cytokines include chemokines. Please rearrange accordingly (lines 41, 46 and so on)

Author Response

Reviewer (2) point

The field of bioinformatics, computational analysis and simulation is attracting the increasing attention due to the accumulating amount of life science data. Indeed, it can significantly reduce the work load and improve efficacy and quality of biomedical research, particularly when numerous compounds have to be analyzed. However, the computation-based approach does not yet considered as completely reliable in terms of practical significance. With that, the major concern is whether the designed in silico experiments can be reproduced during the experimental studies. I would suggest authors to perform additional experiments. For instance, in situ binding capacity of the studied compound and selected cytokines can be tested. 

Thank you honourable reviewer for this comment. It is good suggestion I have discussed with my colleague to working on wet laboratory setup. Actually, I am not experienced with this setup however, we do not have wet lab experiment facilities at this moment just we have only computational lab, in the future we will take account in the next proposal.

Other minor comments are the following: 

1) English grammar checking is required; 

Thank you again for this comment. We further checked the English grammar in the manuscript. We have checked the English grammar of the manuscript by Dr. Sayed Ali

2) Table 1 is not the part of the result and has to be transferred to the introduction; 

Thank you again for this comment. We have removed the table to the introduction section.

3) Are there any improvements of computational analysis software provided by authors? If so, it has to be included as a part of the results; 

Thank you again for this question. We have added 3D representations (Figure 2-3) instead of 2D and transferred the 2D representations to the supplementary file.

4) One of the confirmation tools for the efficacy of the provided analysis can be the comparison with the known compounds with high and low affinity to cytokines (as positive and negative controls); 

Thank you again for this comment. The compound further used for molecular docking analysis to determine their binding energies with different cytokines. The result of the analysis indicated that high binding energies of the most cytokines ranged between − 5.3 and − 7.5 kcal/mol, better than the binding energies of MiP1α (−1.1kcal/mol), IP10 (−4.2kcal/mol), IL-8 (−3.8kcal/mol) and IL-2 (− 4.3 kcal/mol) (Table. 2).

5) In the introduction ("Other names for cytokines are lymphokine (cytokines pro- 27 duced by lymphocytes), monokine (cytokines produced by monocytes), chemokine (cy- 28 tokines with chemotactic activity), and interleukin (cytokines produced by interleukin) 29 (cytokines made by one leukocyte and acting on other leukocytes)") the various cytokine classifications are mixed. Please separate the cell origin and functional cytokines. 

Thank you again for this comment. Other names for cytokines are lymphokine (cytokines produced by lymphocytes), monokine (cytokines produced by monocytes), chemokine (cytokines with chemotactic activity), and interleukin (cytokines produced by interleukin) (cytokines made by one leukocyte and acting on other leukocytes)

6) Cytokines include chemokines. Please rearrange accordingly (lines 41, 46 and so on)

Thank you again for this comment. We followed the following arrangement:

Cytokines: IL-1, IL-1RA, IL-7, IL-9, IL-10, FGF, G-CSF, GM-CSF, PDGF, VEGF, IFN, TNF

Chemokines: CXCL8, IP10, MCP1, MIP1, MIP1

Round 2

Reviewer 1 Report

I want to thank the author for the significant improvement in the manuscript.
- Kindly remove these proteins from the manuscript (2B8U, 1IL8, 2GMF, 1O7Y, 1HMC, 2X6G, 4RAL). These proteins do not contain a co-crystallized ligand or even a natural substrate.
I have highlighted this issue previously (Any protein that does not contain a co-crystallized ligand should be removed from the docking experiment as a result of lack of binding evidence).

- Kindly transfer this sentence to the introduction: "We selected eighteen cytokines for this study [26]. Their classification, main sources, receptor, target cell and major function is listed in Table 1."

- In the "Rationale of the Selected Cytokines" section, kindly state that you chose 18 cytokines for the study. However, you did not include (2B8U, 1IL8, 2GMF, 1O7Y, 1HMC, 2X6G, and 4RAL) in the docking study as there is no information (natural substrate) or reference (native ligand) to compare Andrographolide. 

Thank you.

Author Response

Reviewer - 1

I want to thank the author for the significant improvement in the manuscript.
- Kindly remove these proteins from the manuscript (2B8U, 1IL8, 2GMF, 1O7Y, 1HMC, 2X6G, 4RAL). These proteins do not contain a co-crystallized ligand or even a natural substrate.
I have highlighted this issue previously (Any protein that does not contain a co-crystallized ligand should be removed from the docking experiment as a result of lack of binding evidence).

Answer: Thank you honourable reviewer for this comment. As the above-mentioned proteins do not contain co-crystallized ligand, we have validated the binding affinity protocol by docking Compound 52 and reparixin with IL-4 and IL-8 respectively. From extensive literature review, we found that Compound 52 and reparixin are inhibitors of IL-4 and IL-8 cytokines. We performed molecular docking with the same parameters and software and found that compound 52 and reparixin exhibit binding affinity of -5.3 kcal/mol and -5.8 kcal/mol with the 2 cytokines. We mentioned this in the manuscript and added the interactions figure in the supplementary file.

- Kindly transfer this sentence to the introduction: "We selected eighteen cytokines for this study [26]. Their classification, main sources, receptor, target cell and major function is listed in Table 1."

Thank you honourable reviewer for this comments, I have transfer it.

- In the "Rationale of the Selected Cytokines" section, kindly state that you chose 18 cytokines for the study. However, you did not include (2B8U, 1IL8, 2GMF, 1O7Y, 1HMC, 2X6G, and 4RAL) in the docking study as there is no information (natural substrate) or reference (native ligand) to compare Andrographolide. 

Thank you again for this comment. We have addressed this issue in the previous questions.

Reviewer 2 Report

Dear authors, 

thank you for resolving the stated questions. However, some of the concerns still remain and are required to be fulfilled to strengthen the manuscript. 

The question of cytokine classification remained not fully addressed by the authors. Of note, cytokines include chemokines, and is not completely correct to oppose cytokines and chemokines. Precisely, cytokines are IL-1, IL-1RA, IL-7, IL-9, IL-10, FGF, G-CSF, GM-CSF, PDGF, VEGF, IFN, TNF, CXCL8, IP10, MCP1, MIP1, MIP1. And chemokines (CCL2, CXCL10, etc) are a subfamily of cytokines. 

Another misinterpretation is the classification of cytokines according to the cellular source. I agree that different cells produce diversified cytokine spectrum. However, chemokines and interleukins are off topic in this classification, as they can be produced by all cell types. I recommend the authors to limit this statement with lymphokines and monokines. Moreover, interleukins themselves are not producers of cytokines - interleukins are cytokines themselves. 

I would recommend to double-check this paragraph and further improve it. 

As for point 4, the question was slightly different from that addressed by the authors. Is it still possible to check other compounds, which cytokine binding activity is known? In this case the software efficiency can be validated. Otherwise, we do not have enough information to predict the cytokine binding activities with the new method suggested by the authors. 

Author Response

Reviewer - 2

thank you for resolving the stated questions. However, some of the concerns still remain and are required to be fulfilled to strengthen the manuscript. 

The question of cytokine classification remained not fully addressed by the authors. Of note, cytokines include chemokines, and is not completely correct to oppose cytokines and chemokines. Precisely, cytokines are IL-1, IL-1RA, IL-7, IL-9, IL-10, FGF, G-CSF, GM-CSF, PDGF, VEGF, IFN, TNF, CXCL8, IP10, MCP1, MIP1, MIP1. And chemokines (CCL2, CXCL10, etc) are a subfamily of cytokines. 

Another misinterpretation is the classification of cytokines according to the cellular source. I agree that different cells produce diversified cytokine spectrum. However, chemokines and interleukins are off topic in this classification, as they can be produced by all cell types. I recommend the authors to limit this statement with lymphokines and monokines. Moreover, interleukins themselves are not producers of cytokines - interleukins are cytokines themselves. 

I would recommend to double-check this paragraph and further improve it. 

Cytokines are small proteins secreted by cells , These are classified as mononuclear cells such as macrophages have been referred to as monokines while the cytokines produced by activated T lymphocytes are termed lymphokines.It can be classified cytokines according split of labor , T-helpor -1 of cytokines such as interleukin (IL)-2, IL-12 , TNF-a, IFN-γ ,T helper -2 of cytokiens such as IL-3, IL-4, IL-5 , IL-13  and T-helper-3 of cytokines such as IL-10 , TGF-β. Chemokines like CCL2 and CXCL10 are a small cytokines, or called signalling proteins secreted by cells. It has ability to induce directed chemotaxis in nearby responsive cells; they are chemotactic cytokines. They are secreted as result of inflammatory response.

As for point 4, the question was slightly different from that addressed by the authors. Is it still possible to check other compounds, which cytokine binding activity is known? In this case the software efficiency can be validated. Otherwise, we do not have enough information to predict the cytokine binding activities with the new method suggested by the authors. 

Answer: Thank you honourable reviewer for this comment. We have validated the software efficiency by by docking Compound 52 and reparixin with IL-4 and IL-8 respectively. From extensive literature review, we found that Compound 52 and reparixin are inhibitors of IL-4 and IL-8 cytokines. We performed molecular docking with the same parameters and software and found that compound 52 and reparixin exhibit binding affinity of -5.3 kcal/mol and -5.8 kcal/mol with the 2 cytokines. We mentioned this in the manuscript and added the interactions figure in the supplementary file.
